# Metabolic diversity within the globally abundant Marine Group II Euryarchaea offers insight into ecological patterns

Benjamin J. Tully [1,2]

Despite their discovery over 25 years ago, the Marine Group II Euryarchaea (MGII) remain a difficult group of organisms to study, lacking cultured isolates and genome references. The MGII have been identified in marine samples from around the world, and evidence supports a photoheterotrophic lifestyle combining phototrophy via proteorhodopsins with the remineralization of high molecular weight organic matter. Divided between two clades, the MGII have distinct ecological patterns that are not understood based on the limited number of available genomes. Here, I present a comparative genomic analysis of 250 MGII genomes, providing a comprehensive investigation of these mesophilic archaea. This analysis identifies 17 distinct subclades including nine subclades that previously lacked reference genomes. The metabolic potential and distribution of the MGII genera reveals distinct roles in the environment, identifying algal-saccharide-degrading coastal subclades, protein-degrading oligotrophic surface ocean subclades, and mesopelagic subclades lacking proteorhodopsins, common in all other subclades.

[1] Department of Biological Sciences, University of Southern California, Los Angeles, CA 90089, USA. [2] Center for Dark Energy Biosphere Investigations, University of Southern California, Los Angeles, CA 90089, USA. Correspondence and requests for materials should be addressed to B.J.T. (email: tully.bj@gmail.com)

Since their discovery by DeLong[1] in 1992, despite global distribution and representing a significant portion of the microbial plankton in the photic zone, the Marine Group II Euryarchaea (MGII) have remained an enigmatic group of organisms in the marine environment. The MGII have been identified at high abundance in surface oceans[2,3] and can account for ~15% of the archaeal cells in the oligotrophic open ocean[4]. The MGII have been shown to increase in abundance in response to phytoplankton blooms[5] and can comprise up to ~30% of the total microbial community after a bloom terminates[6]. Research has shown that the MGII correlate with specific genera of phytoplankton[7], during and after blooms[8], can be associated with particles when samples are size fractionated[9], and correlate with a novel clade of marine viruses[10]. Phylogenetic analyses have revealed the presence of two dominant clades of MGII, referred to as MGIIa and MGIIb (recently Thalassoarchaea has been proposed as a name for the MGIIb[11]), that respond to different environmental conditions, including temperature and nutrients[12].

To date, the MGII have not been successfully cultured or enriched from the marine environment. Instead our current understanding of the role these organisms play is derived from interpretations of environmental sampling data (e.g., phytoplankton- and particle-associated) and a limited number of genomic fragments and reconstructed environmental genomes. Collectively, these genomic studies have revealed a number of re-occurring traits common to the MGII, including: proteorhodopsins in MGII sampled from the photic zone[13], genes targeting the degradation of high molecular weight (HMW) organic matter, such as proteins, carbohydrates, and lipids, and subsequent transport of constituent components into the cell[11,14–16], genes representative of particle-attachment[9,14], and genes for the biosynthesis of tetraether lipids[11,17]. Comparatively, the capacity for motility via archaeal flagellum has only been identified in some of the recovered genomes[11,14]. Much of this primary literature is reviewed in ref. [18].

The global prevalence of the MGII and their predicted role in HMW organic matter degradation make them a crucial group of organisms for understanding remineralization in the global ocean. Evidence supports specialization of MGIIa and MGIIb to certain environmental conditions, but the extent of this relationship in the oceans are not understood and cannot be discerned from the available genomic data. The environmental genomes reconstructed from the Tara Oceans metagenomic datasets[19–22] provide an avenue for exploring the metabolic variation between the MGIIa and MGIIb, and in conjunction with environmental data collected from the same filter fractions and sampling depths[23,24] can be used to understand the variables and conditions that favor each clade.

Here, the analysis of 250 non-redundant MGII genomes identifies the metabolic traits unique to the genomes derived from the MGIIa and MGIIb, providing new context for the ecological roles each clade plays in remineralization of HMW organic matter. Further, the MGIIa and MGIIb can be assigned to 17 subclades, with distinct ecological patterns with respect to sample depth, particle size, temperature, and nutrient concentrations.

## Results

**Putative phylogeny of a non-redundant set of MGII genomes.** Despite their global abundance and active role in the cycling of organic matter, it has been difficult to glean metabolic information from the MGII. As of January 2018, a total of 20 MGII genomes with sufficient quality metrics (>50% complete and <10% contamination) had been reconstructed from environmental metagenomic data and analyzed[11,14,17,25,26]. This number

could be supplemented with two single-amplified genomes (SAGs) accessed from JGI that were determined to be ~40% complete, but possessed 16S rRNA gene sequences. These publicly available genomes were severely skewed towards the MGIIb[17,25,26] (16 genomes) with only six genomes for the MGIIa available[14,17,25]. A combined 410 genomes reconstructed from marine environmental metagenomes, originating from four studies utilizing the Tara Oceans dataset (designations TMED[19], TOBG[20], UBA[21], and TARA-MAG[22]) and the Red Sea (designated as REDSEA[27]), were identified in publicly available databases. As the metagenome-assembled genomes generated from studies using the Tara Oceans data were generated from the same metagenomes, it was important to identify potentially identical genomes within the recovered genome collection. Groups of genomes were identified that shared ≥98.5% average nucleotide identity (ANI) and a representative was selected for each group based on completion and contamination metrics. In the past, ~96% ANI has been used to denote organisms of different species[28,29]. A cutoff of 98.5% ANI was selected as to only exclude identical genomes, while attempting to retain genomes with strain level differences sampled from different metagenomes. A total of 90 groups of at least two genomes were identified (Supplementary Data 1) and removal of redundant genomes resulted in a dataset of 258 non-redundant genomes. A phylogenetic tree using 120 concatenated single copy marker proteins was constructed using all genomes that had ≥60 single copy markers (Fig. 1). Eight genomes had an insufficient number of markers and were no longer considered for further analysis (a tree with the same parameters was applied to the redundant genome dataset and included 410 genomes; Supplementary Figure 1; Supplementary Table 1 and 2). The MGIIa and MGIIb formed two distinct branches with a majority of genomes ($n = 205$) belonging to the MGIIb. The genomes further clustered into 17 distinct subclades —8 MGIIa subclades (designated MGIIa.1–8) and 9 MGIIb subclades (designated MGIIb.9–17). The 17 subclades were further supported by pairwise ANI and average amino acid identity (AAI; Supplementary Figure 2 and 3; Supplementary Data 2). Nine of the clades were composed exclusively of genomes reconstructed from the Tara Oceans metagenomic dataset. Based on the extrapolated genome size for these 17 subclades, MGIIa genome sizes were significantly larger than MGIIb genomes, on average ~400 kbp (Fig. 2a; $p_{\text{Student's } t\text{-test}}$ << 0.001). The two most basal clades of the MGIIb have mean genome sizes similar to that of the MGIIA. In contrast, there was no clear relationship between percent GC (%G+C) content and phylogenetic group. %G+C content of the genomes had a wide range of values (~35%−>60%; Fig. 2b). Additionally, several subclades had high internal variation of %G+C content.

A subset of the MGII genomes had 16S rRNA gene sequences ($n = 34$) which were used to determine the relationship between previously identified sequence clusters and the newly identified subclades (Supplementary Figure 4). 16S rRNA sequences from the MGIIa.1 and MGIIb.9 were not represented in previously identified MGII 16S rRNA gene clusters. Conversely, the previously identified N cluster did not have any representative amongst the environmental genomes, either as a result of missing diversity among the described genomes or due to the fact that not all of the subclades had a representative with an identified 16S rRNA gene. Several defined 16S rRNA gene clusters could be linked directly to subclades with genomic representatives based on phylogeny; the WHARN cluster branched with a representative of the MGIIb.15 and the M and L clusters branched with the MGIIa.6 and MGIIa.8, respectively. The O cluster from the MGIIb, could be divided at several internal nodes and assigned to five of the subclades. The phylogeny of several of the representative 16S rRNA sequences associated with the O cluster

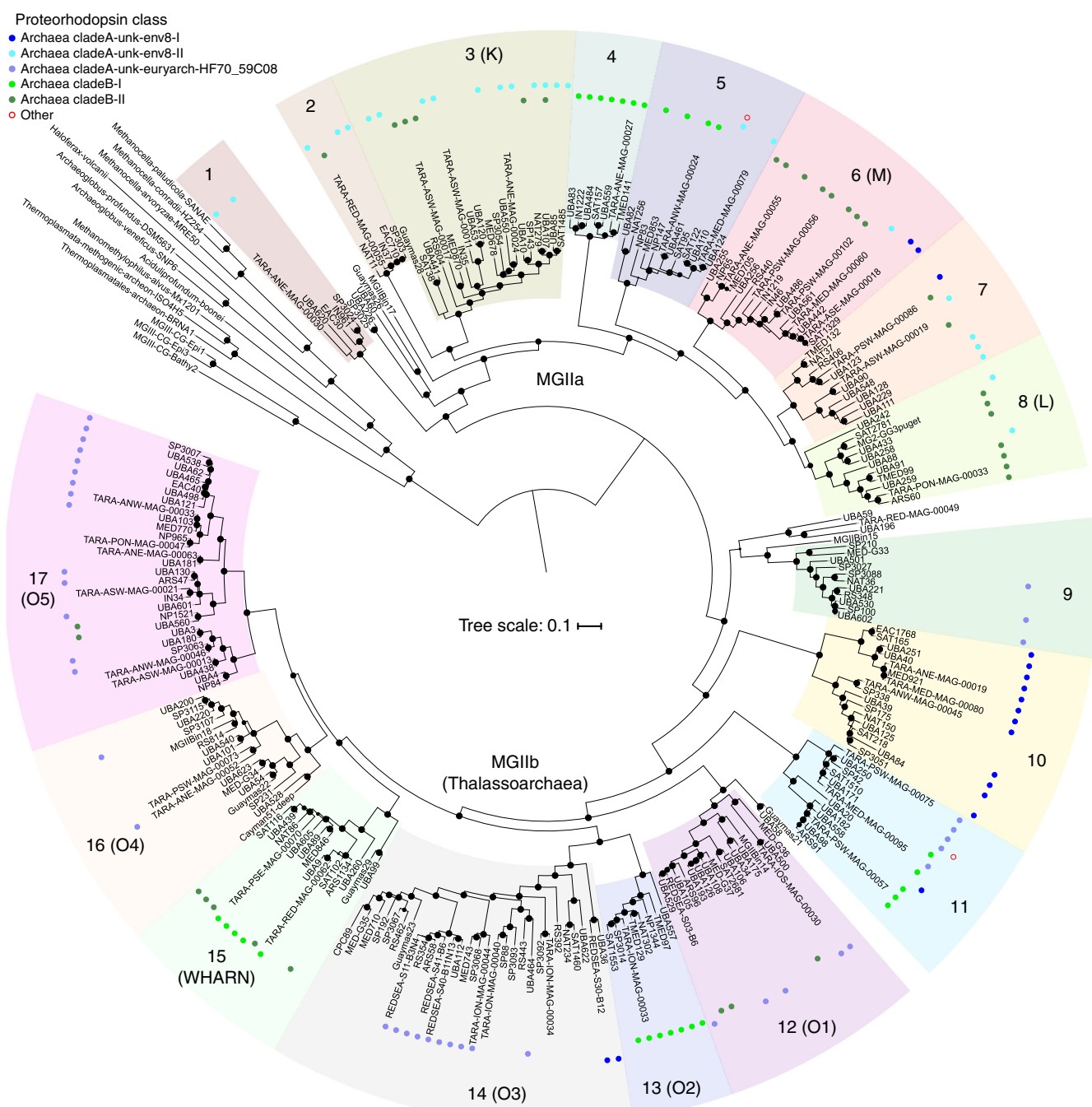

**Fig. 1** A phylogenomic tree for the MGII. Constructed using 120 concatenated single copy marker proteins, subclades are denoted based on relative distance from the root and supported by %ANI and %AAI. Parentheses indicate subclades that can be attributed to previously defined 16S rRNA gene clusters. Bootstrap values are scaled proportionally between 0.75 and 1. Identified proteorhodopsin classes are denoted with colored circles based on the predicted color of light for which the proteorhodopsin is spectrally tuned. Source data are provided as a Source Data file

could not be fully resolved likely as the result of a low quality sequence across the length of the alignment or due to an incorrect placement within the genome during binning.

**An electron transport chain with putative Na⁺ pumping components**. There were several shared traits amongst the MGIIa and MGIIb, particularly related to the components of the electron transport chain (ETC). Genomes belonging to both groups had canonical NADH dehydrogenases (complex I) and succinate dehydrogenases (complex II) that link electron transport to oxygen as a terminal electron acceptor via low-affinity

cytochrome c oxidases (Fig. 3). As has been noted previously[9], most members of the MGII possessed genes encoding a cytochrome b and a Rieske iron–sulfur domain protein but lacked the genes for the canonical cytochrome $bc_1$ (complex III). Many of the MGII families also possessed RnfB, an iron–sulfur protein that can accept electrons from ferredoxin and transfer them to the ETC. The complete Rnf complex is capable of generating a Na⁺ gradient through the oxidation of ferredoxin but all members of MGII lacked the subunits needed to complete the complex (RnfACDEG). Thus, it was surprising that distributed across all of the subclades in 188 genomes (75%), the MGII possessed an $A_1A_O$ ATP synthase that, based on the presence of specific motifs

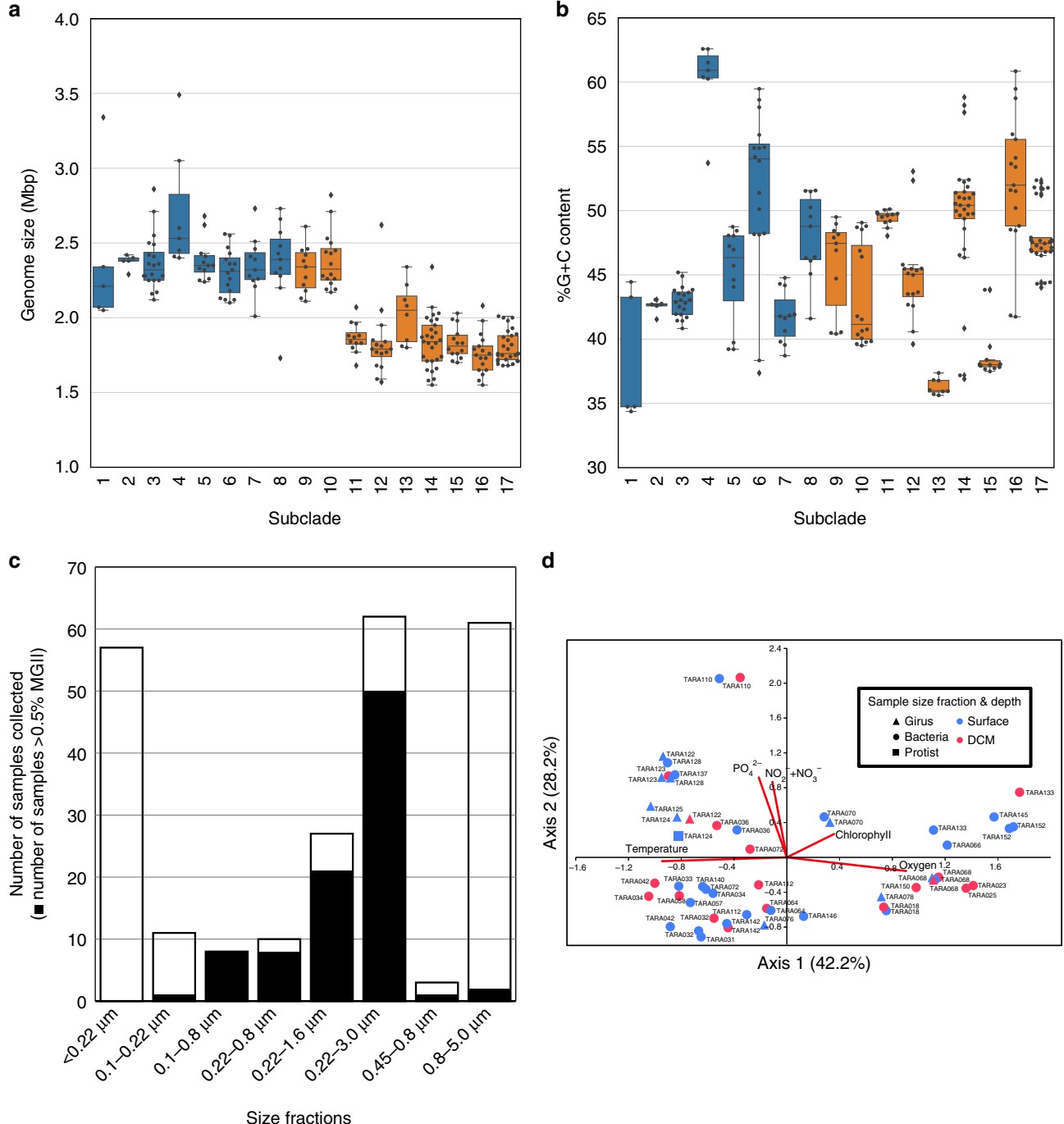

**Fig. 2** The relationships between the MGII and their environment. **a** Distribution of genome sizes for each subclade (center line, median; box limits, upper and lower quartiles; whiskers, 1.5×interquartile range; diamonds, outliers). MGIIa and MGIIb box plots are displayed in blue and orange, respectively. **b** Distribution of genome %G+C content for each subclade (error bars as in Fig. 2A). MGIIa and MGIIb box plots are displayed in blue and orange, respectively. **c** The outline illustrates the number of metagenomic samples available from the *Tara* Oceans dataset for a given filter fraction and solid filled portion represents the number of samples in that size fraction that recruit ≥0.5% of metagenomic reads to the MGII. **d** A canonical correspondence analysis (CCA) of the high abundance (≥0.5% relative fraction) samples (*n* = 54) based on the RPKM (reads per kbp of each genome per Mbp of each metagenomic sample) values for the MGII genomes. Blue and red symbols denote surface and deep chlorophyll maximum samples, respectively. Triangles, circles, and squares denote the girus (0.22–0.8 µm), bacteria (0.22–1.6 µm), and protist (0.8–5.0 µm) *Tara* Oceans size fractions, respectively. Source data are provided as a Source Data file

in the c ring protein (AtpK), could be inferred to generate ATP through the pumping of $Na^+$ ions. All of the genomes had the necessary conserved glutamine and a motif in respective transmembrane helices[30] (Supplementary Figure 5A). The motif in the second helix appears to be diagnostic of the clade a genome belongs to: the MGIIa contained a LPESxxI motif and the MGIIb contained a LPETIxL motif. The presence of these motifs does not preclude ATP synthesis via $H^+$ pumping[31], though a majority of the experimentally confirmed $A_1A_O$ ATP synthases with these motifs exclusively pump $Na^+$ ions[30].

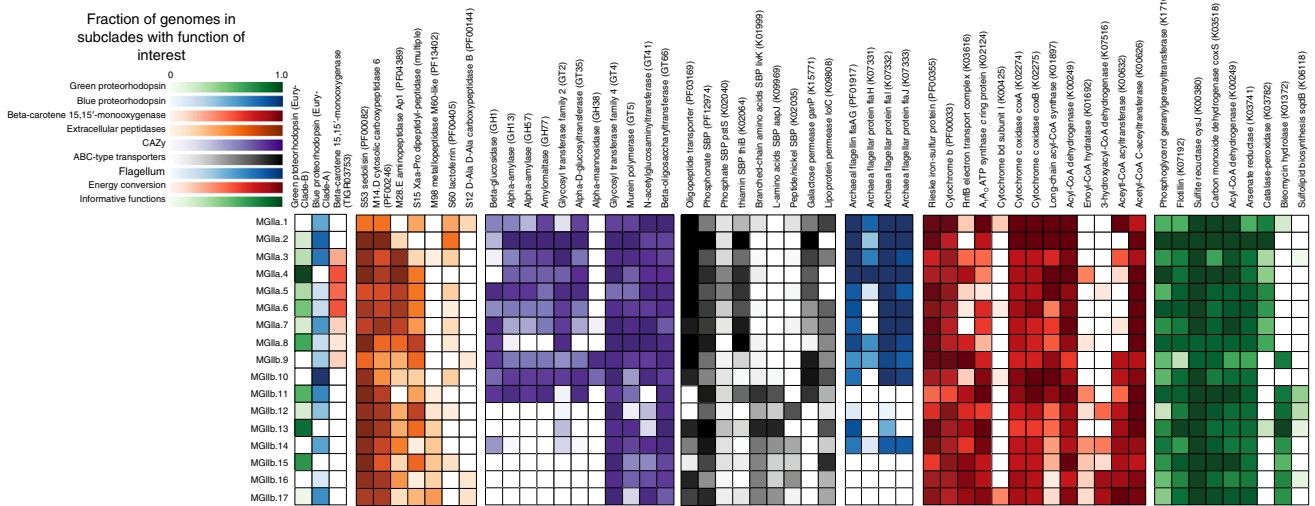

**Fig. 3** Occurrence of various functions of interest. Heatmaps are scaled from 0 to 1, where 1 represents that all genomes within the designated subclade possess the function of interest. SBP, substrate-binding protein. Source data are provided as a Source Data file

**Shared ability to degrade extracellular proteins and fatty acids.** As has been reported previously[11,14–16], a majority of the MGII families are poised to exploit HMW organic matter. The families share the potential to degrade and import proteinous material with two extracellular peptidases (sedolisin-like peptidases and carboxypeptidase subfamily M14D) and an oligopeptide transporter present in most of genomes (Fig. 3). All of the MGII families appear capable of some degree of fatty acid degradation due to the presence of acyl-CoA dehydrogenase and acetyl-CoA C-acetyltransferase, though some of the intermediate steps are missing from all genomes in several families (Fig. 3). It is unclear if the incomplete nature of the pathway in these families is the result of uncharacterized family-specific analogs or some degree of metabolic hand-off between different organisms degrading fatty acids. Several other metabolic traits that had been reported in genomes belonging to either the MGIIa or MGIIb are also part of the MGII core genome[11,17], including the capacity for the assimilatory reduction of sulfite to sulfide, the transport of phosphonates, flotillin-like proteins, which may have a role in cell adhesion, and geranylgeranylglyceryl phosphate (GGGP) synthase, a key gene for tetraether lipid biosynthesis (Fig. 3).

**Putative proteorhodopsins differentiate members of the MGII.** Components of the ETC and HMW degradation were present in all the subclades and there were several traits that either lacked a phylogenetic signature or differentiated the MGIIa and the MGIIb. As has been noted previously[14] and confirmed with this collection of genomes, taxa in all of the MGII subclades possess genes encoding light-sensing rhodopsins and, based on the amino acids at positions 97 (aspartate) and 108 (lysine/glutamic acid) in the rhodopsin sequences, are predicted to function as proteorhodopsins (PRs)[32] capable of establishing H$^+$ gradients (Supplementary Figure 5B). Phylogenetically, these PRs cluster in established clades[33] Archaea Clade A (Clade-A) and Archaea Clade B (Clade-B) and based on the amino acid in position 105[34] (glutamine/methionine), spectral tuning[35] prediction indicates sensitivity to blue and green light, respectively (Supplementary Figure 5B). Five subclades exclusively possess Clade-A, three subclades exclusively possess Clade-B, and nine subclades have genomes that possess either of the two PRs.

A group of genomes from MGIIb.16, including a number of genomes reconstructed or present (see below) in the aphotic zone,

do not possess PRs (Fig. 1). The lack of PRs in deep-sea MGII is consistent across the tree, with deep-sea reconstructed genomes not within the MGIIb.16 tending to represent the most basal branching members of other subclades (e.g., genome Guaymas21 basal to MGIIb.12). PRs from Clade-A fall into three distinct phylogenetic groups associated with the clades unk-env8 (CladeA-unk-env8-I and -II) and unk-euryarch-HF70_59C08 as identified in the MICrhoDE database, while Clade-B has two distinct groups (Clade-B-I and -II) (Supplementary Figure 6). The MGIIa possessed all of the PR groups, except unk-euryarch-HF70_59C08 and slightly favor the green light tuned PRs (55% of PR containing genomes), while the MGIIb do not utilize the CladeA-unk-env8-II group and favor blue-light tuned PRs (67% of PR containing genomes). Additionally, several subclades possessed exclusively one of the PR clades (Fig. 1). Despite the requirement of the chromophore retinal for the functioning of PR, a majority of the MGII lacked an annotation for beta-carotene 15,15′-monooxygenase (Fig. 3), essential for the last cleavage step needed to activate retinal. Two of the eight subclades from the MGIIa and all but one of the subclades from the MGIIb lacked this crucial functional step.

**Degradation of proteins and oligosaccharides differentiate MGII.** Whereas the MGII shared several functionalities with a role in the degradation of HMW organic matter, there was a greater diversity of functionality in specific clades and subclades (Fig. 3). There were five additional classes of extracellular peptidases (aminopeptidases subfamily M28E, dipeptidyl-peptidase, M60-like metallopeptidase, lactoferrin-like, and carboxypeptidase B) common (and 19 extracellular peptidases with infrequent occurrence; Supplementary Data 3) amongst the genomes. The collective suite of peptidases within a genome dictate the potential types of proteinous material that can be processed by an organism. Three of the five extracellular peptidase classes were distributed across both the MGIIa and MGIIb, while the M60-like metallopeptidase and carboxypeptidase B, were present almost exclusively amongst the MGIIb. Despite sharing many of the putative protein-degrading functions, subclades from the MGIIb, except for MGIIb.9 and MGIIb.10, possess the substrate-binding proteins for ATP-binding cassette (ABC) type transporters for three additional amino acid and peptide transporters (branched-chain amino acids, L-amino acids, and peptide/nickel), while the

MGIIa only have the previously noted oligopeptide transporter (Fig. 3).

Beyond the degradation of proteins and fatty acids, there is evidence to suggest that MGII have a role in the degradation of carbohydrate HMW organic matter[36]. Interestingly, glycoside hydrolases with functionality for the degradation of algal oligosaccharides, including pectin, starch, and glycogen, are found exclusively amongst the MGIIa and the most basal subclades of the MGIIb, the MGIIb.9, MGIIb.10, and MGIIb.11 (Fig. 3). These same subclades also possess an annotated galactose permease subunit for an ABC-type transporter. Further, MGIIb.9 and MGIIb.10 also possess a glycoside hydrolase that could possibly play a role in mannosylglycerate degradation, an osmolyte found in red algae[37].

**Motility is a trait common to the MGIIa.** Previous research has shown evidence for and against the putative capacity for motility amongst the MGII[11,14]. The analyzed genomes lacked annotations or homology for most of the canonical archaeal flagellum operon (Fig. 4; Supplementary Table 3). However, genomes from all of the MGIIa subclades, MGIIb.9, MGIIb.10, and MGIIb.14 possessed proteins annotated as subunits from the canonical operon (FlaAGHIJ). A comparison of the identified subunits from a representative of the MGIIa.1 to *Methanococcus voltae* A3 revealed 40–70% amino acid similarity between putative orthologs. These subunits were syntenic in a region that contained an additional 1–3 identifiable flagellins and several orthologous proteins lacking annotations. All of the predicted proteins in this region could be identified by a similarity comparison between representatives of each family. The structure of the region, including the predicted proteins immediately up- and downstream of the region, appeared to be mostly conserved amongst the MGIIa, while some variation in gene content could be observed amongst the subclades from the MGIIb.

For several other functions ascribed to the MGII as a whole[11], there are distinct distributions amongst the clades, including the presence of a catalase-peroxidase amongst the MGIIa and a bleomycin hydrolase amongst the MGIIb (Fig. 3). Further, several other predicted metabolic functions appear to be specific to only a subset of subclades and may have a role in niche differentiation, including cytochrome bd (a high-affinity oxygen cytochrome responsible for microaerobic respiration), a phosphate substrate-binding subunit for an ABC-type transporter, and UDP-sulfoquinovose synthase, a key gene for the biosynthesis of sulfolipids (Fig. 3).

**Subclades belong to ecological clusters with unique distributions.** Using a comprehensive set of *Tara* Oceans metagenomic datasets from across the globe[24], that included all of the size fractions for which DNA was collected (viral, 'bacterial', and eukaryotic), it was possible to explore where specific MGII groups were dominant. The MGII were rarely found to be abundant (>0.5% relative abundance; mean, 1.75%; maximum, 4.97%) in samples for size fractions <0.22 μm or >0.8 μm, with almost all abundant samples occurring in the 'bacterial' size fractions (0.1–3.0 μm; Fig. 2c). Globally, the MGII were abundant at all *Tara* Oceans stations with a 'bacterial' size fraction ($n = 45$), except for six stations (Supplementary Figure 7). There were no *Tara* Oceans metagenomic samples collected from size fractions >5 μm. A canonical correspondence analysis (CCA) based on the MGII community structure for 54 samples divided samples along a gradient of temperature and oxygen concentrations (x-axis, 42.2% observed variance) and on nutrient concentrations (y-axis, 28.2% observed variance; Fig. 2d). Overall, samples from the surface and deep chlorophyll maximum (DCM) were intermingled in all quadrants of the CCA plot suggesting that incidence light availability may not be a contributing factor to community composition. Interestingly, while the MGII are not persistently abundant in the smallest *Tara* Oceans size fractions, of the 11 girus size fraction samples on the CCA plot, a majority (55%) group together correlated to high temperature and phosphate and nitrate concentrations.

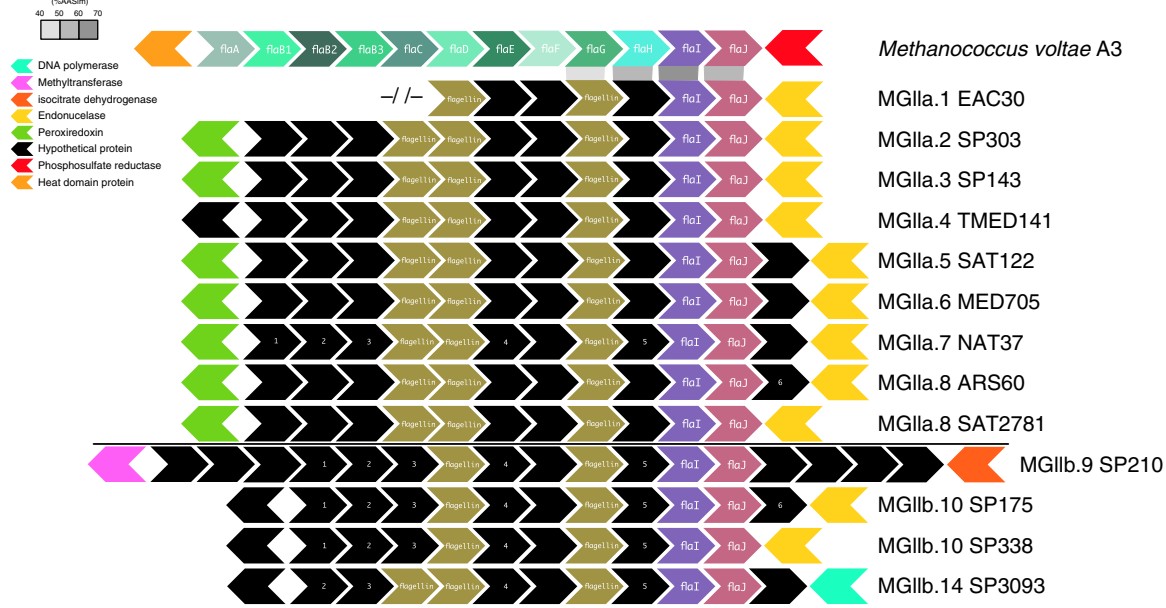

**Fig. 4** Putative MGII motility operon. The longest contig for each subclade is shown. Hypothetical proteins lacking KEGG annotations or matches to queried HMMs are in black. All hypothetical proteins in a column had significant BLAST matches to their neighbors, except as noted by the numbers in the transition from the MGIIa to the MGIIb. Flagellins noted in the gold segment were detected using the archaeal flagellin PFAM (PF01917). Proteins immediately upstream and downstream are colored based on predicted function. Significant BLAST matches between *Methanococcus voltae* A3 and the MGIIa.1 genome are noted. Source data are provided as a Source Data file

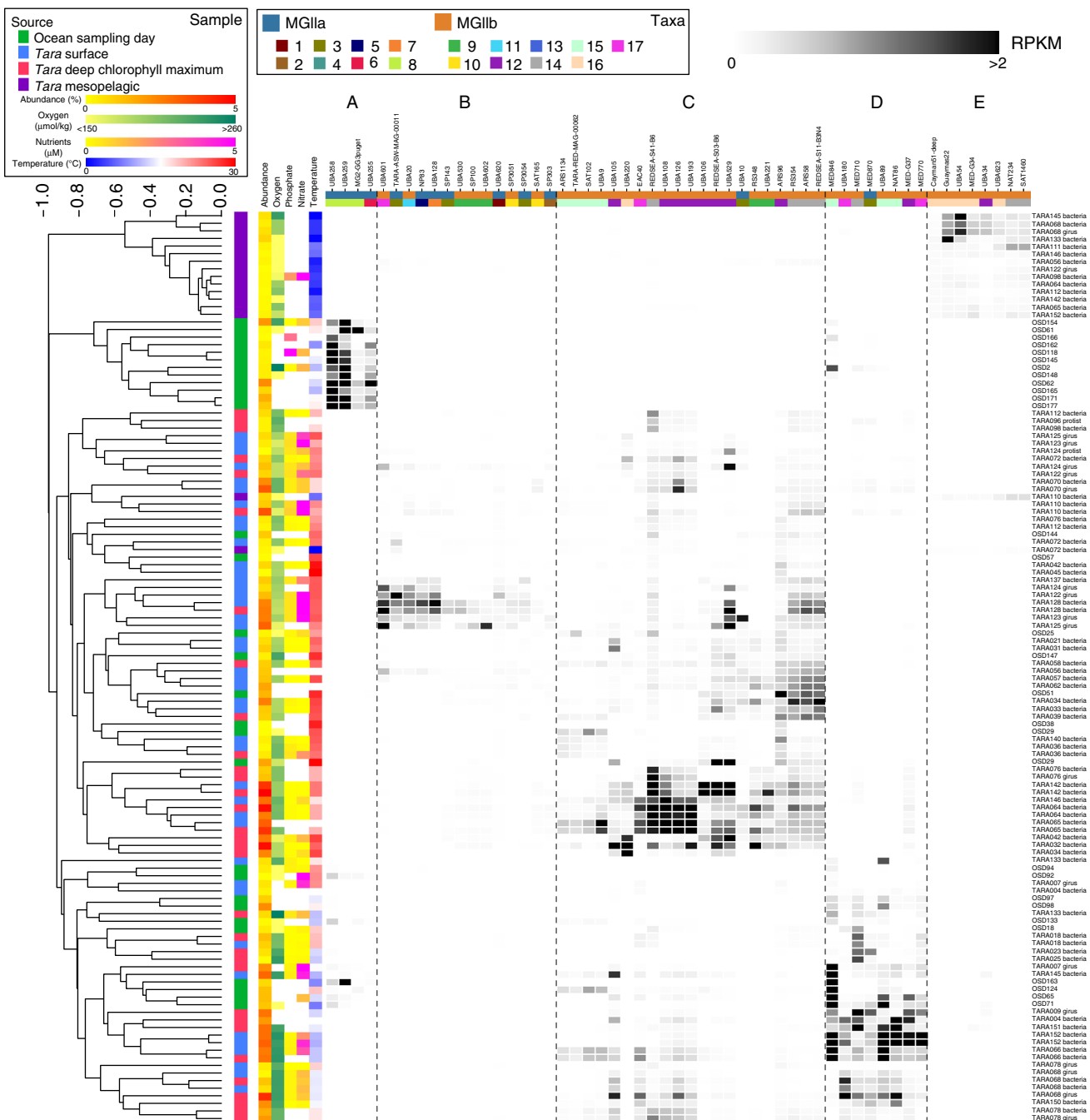

**Fig. 5** MGII global relative abundance. RPKM values for a subset of MGII genomes analyzed from high abundance samples (≥0.5% relative fraction). RPKM values are scaled from 0 to 2 with values ≥2 in black (median, 0.0012; maximum, 13.09). Samples are hierarchically clustered based on all MGII RPKM values and the sample source is displayed. The available environmental parameters are presented as colored heatmaps (missing parameters are not displayed). The heatmap used to identify high abundance clusters is available in Supplementary Figure 8. Source data are provided as a Source Data file

The abundance of the MGIIa in open ocean samples was limited. In an effort to identify samples where the clade may be abundant, 118 'prokaryotic' metagenomes from coastal (<10 km) Ocean Sampling Day[38] 2014 (OSD) samples were assessed for the presence of the MGII genomes (Fig. 5; Supplementary Figure 9). These samples were collected using a unified method that captured whole seawater >0.22 μm and measured a limited number of physical properties, generally, temperature, salinity, distance to the coast, and depth (0–5 m). Unlike the ubiquitous nature of MGII in the 'bacterial' *Tara* Oceans fractions, only about a quarter of the samples (n = 31) from OSD had high MGII abundance.

*Tara* Oceans and OSD samples and taxa were hierarchically clustered based on community composition using Bray-Curtis distances. Five groups of taxa that occurred at high abundance in a discreet set of samples were identified (Supplementary Figure 8). These groups were classified as "Ecological clusters" A–E, an indication of the shared ecology between the taxa. Ecological cluster A consisted of a group of four MGIIa, predominantly from MGIIa.8, that were abundant exclusively in samples from the OSD (Fig. 5). Though coastal, these OSD samples were marine in nature (30–38 practical salinity unit [PSU] compared to *Tara* Oceans samples 29–40 PSU). The lack of consistency in the types of data collected from OSD samples limits the statistical power for

interpreting the influence of oxygen and nutrients concentration on these taxa ($p_{BH-FDR} < 0.02$), but these OSD samples have moderate temperatures (12–21 °C) compared to many of the other samples ($F_{PERMANOVA} = 12.04$, $p_{BH-FDR} = 0.0001$). Ecological cluster E consisted of a group of taxa that were exclusively abundant in mesopelagic samples. Though the cluster does exclude UBA528, which is the most abundant taxa in all mesopelagic samples (Supplementary Figure 8). This cluster is entirely composed of taxa from the MGIIb, with a majority from the MGIIb.16. All but one mesopelagic sample lack nutrient data, but Ecological cluster E was significantly correlated to low oxygen concentrations (3–237 µmol/kg, $F_{PERMANOVA} = 2.884$, $p_{BH-FDR} = 0.0002$) and low temperatures (<10.8 °C, $F_{PERMANOVA} = 7.95$, $p_{BH-FDR} = 0.0002$).

Ecological clusters B–D were less confined to a single sample type, but correlated with specific environmental conditions. Ecological cluster B consisted of 14 taxa from both the MGIIa ($n = 7$) and MGIIb ($n = 7$) that were abundant predominantly in surface samples with warm temperatures (24–27 °C, $F_{PERMANOVA} = 9.559$, $p_{BH-FDR} = 0.0002$), moderate oxygen (160–195 µmol/kg, $F_{PERMANOVA} = 4.315$, $p_{BH-FDR} = 0.0002$), low phosphate (0.5–0.6 µM, $F_{PERMANOVA} = 2.872$, $p_{BH-FDR} = 0.0003$), and high nitrate (2.4–5.5 µM, $F_{PERMANOVA} = 2.003$, $p_{BH-FDR} = 0.0015$) concentrations. Ecological cluster C consists of 21 taxa almost exclusively from the MGIIb ($n_{MGIIb} = 20$) with high abundance in samples from both the surface and DCM, though members of cluster are distributed among all of the high temperature samples (19–31 °C, $F_{PERMANOVA} = 7.579$, $p_{BH-FDR} = 0.0002$). Many of these samples also possessed moderate oxygen (171–212 µmol/kg, $F_{PERMANOVA} = 3.564$, $p_{BH-FDR} = 0.0002$), low phosphate (<0.5 µM, $F_{PERMANOVA} = 1.918$, $p_{BH-FDR} = 0.003$), and low nitrate (<2.1 µM, $F_{PERMANOVA} = 1.38$, $p_{BH-FDR} = 0.0491$) concentrations. Ecological cluster D was predominantly MGIIb taxa ($n_{MGIIb} = 7$) from samples with moderate temperatures (11–21 °C, $F_{PERMANOVA} = 11.36$, $p_{BH-FDR} = 0.0002$), high oxygen (>231 µmol/kg, $F_{PERMANOVA} = 4.99$, $p_{BH-FDR} = 0.0002$), and low phosphate (<0.4 µM, $F_{PERMANOVA} = 1.714$, $p_{BH-FDR} = 0.0212$) concentrations. Ecological cluster D was not statistically correlated with nitrate concentrations.

## Discussion

The details provided by the increased resolution of MGII genomes collected for this study in phylogeny, metabolism, and ecology redefines what is understood about these globally dominant mesophilic Euryarchaea. Previous phylogenetic diversity contained within reconstructed genomes and genomic fragments failed to capture at least nine newly identified subclades. This collection of 250 genomes allows for a more precise understanding of the metabolic potential present in the MGII, including the metabolic and ecological differentiation of the MGIIa and MGIIb.

There are several unifying elements in the overall predicted MGII metabolism. All MGII have the metabolic potential to be obligate aerobic heterotrophs with many possessing the capacity to harness solar energy via PRs. There is no collective indication that the MGII have the capacity for anaerobic metabolisms, though members of four subclades (MGIIa.1, MGIIa.6, MGIIa.10, and MGIIb.17) have the genomic potential to function in suboxic conditions through the presence of a high-affinity cytochrome bd oxidase. Central to the potential heterotrophic metabolism is the degradation of HMW proteins and fatty acids that can be exploited by a number of transporters and extracellular peptidases. Energy conversion via oxidative phosphorylation uses a number of canonical (complexes I and II) and MGII specific (complex III) electron transfer proteins, but the MGII may also be capable of utilizing a sodium motive force through an $A_1A_O$ ATP

synthase. A $Na^+$-driven ATP synthases differentiates the MGII from most marine bacteria that possess $F_1F_O$ $H^+$-driven ATP synthases. How this MGII $H^+/Na^+$ ETC would function in situ is unclear but may be linked to the only identifiable component of the Rnf sodium translocating complex, RnfB. It may be that the MGII utilize both $H^+$ and $Na^+$, similar to *Methanosarcinales* under marine conditions[31], and that different elements of the MGII ETC perform these translocations.

The presence of PRs is ubiquitous across the MGII, with members of the MGIIa and MGIIb predicted to have spectral tuning to both blue (Clade-A) and green (Clade-B) light. But while neither PR clade adheres to a strict MGII phylogenetic distribution, there are specific patterns that may provide insight in to the adaptation of certain subclades or taxa to localized conditions. For example, MGIIa.4, MGIIb.13, and MGIIb.15 possess only green spectral tuned PRs. While taxa from MGIIa.4 and MGIIb.13 individually do not reach high relative abundance, abundant taxa from MGIIb.15 are present in Ecological clusters C and D and are correlated to open ocean samples with distinct temperature and oxygen profiles. The taxa abundant in the two Ecological clusters are phylogenetically distinct within the MGIIb.15 subclade, suggesting fine-scale genomic adaptation may play a role how these organisms are distributed in the ocean. Further, MGII genomes reconstructed from the bathypelagic (Guaymas-derived genomes), abyssopelagic (Cayman-derived genome), and identified as abundant in mesopelagic *Tara* Oceans samples (Ecological cluster E) all lack PRs indicating that MGII heterotrophy is not dependent on light-amended proton motive force.

There are two important metabolic distinctions between the MGIIa and most of the MGIIb—the potential for the degradation of oligosaccharides and flagellum-based motility. Reported in the analysis of the first MGII genome[14], the potential for oligosaccharide degradation has often been ascribed to all members of the MGII. Analysis of the presence of glycoside hydrolases in MGII genomes suggests that this metabolism is limited to the MGIIa and the three most basal subclades of the MGIIb. Further, the specificity of the annotated glycoside hydrolases suggests that the MGII are degrading algal derived substrates. With the exception of MGIIb.11, the subclades with the capacity to degrade algal oligosaccharides also have the potential for motility via a proposed archaeal flagellum-based system. The capacity for motility may be required to exploit ephemeral sources of oligosaccharides through chemotaxis between algal cells/detritus. This metabolic link to algal oligosaccharides may limit the abundance of these subclades to coastal environments, such as MGIIa subclades in Ecological cluster A, or during/post phytoplankton blooms, as previously observed[39].

Many of the individual taxa identified in this collection of 250 genomes are not abundant in any one sample, but the aggregated MGII community dynamics revealed the importance of temperature and oxygen availability to shape the individual constituents of a sample. Both temperature and oxygen availability directly impact the heterotrophic metabolisms of the MGII through the requirement of oxygen by aerobes for carbon degradation and the role temperature plays in enhancing reaction kinetics at warmer temperatures. Combined with measurements for phosphate, nitrate, and chlorophyll, these components of the environment explained more than two-thirds (70%) of the community variance. While chlorophyll measurements capture one of the important biotic interactions, it is likely that the missing variance can/will be explained by MGII-microbe and MGII-virus interactions.

The predicted metabolic functions observed in the subclades appear to correlate with observations of MGII taxa and community constituents in global samples. The most abundant MGIIa

genomes occur in coastal samples originating from OSD samples and not the open ocean *Tara* metagenomes. These coastal conditions would correlate to areas of the oceans where algal oligosaccharides would be more readily available. Additionally, high nitrate concentrations can stimulate growth of phytoplankton, which may explain why Ecological cluster B has a high MGIIa abundance from open ocean samples collected in the Equatorial Pacific Ocean (TARA122–125 and TARA128). Conversely, the MGIIb dominant ecological clusters occur in environments where direct algal inputs are limited (i.e., open ocean and mesopelagic samples). There are several ecological clusters for which the current metabolic predictions cannot explain. Difficulty in this regard is partially a result of the limited degree of functional annotation achieved for the MGII (~45% KEGG functional annotation efficiency). Ecological clusters C and D are composed predominantly of MGIIb subclades and have unique distributions based on non-overlapping temperature and oxygen concentrations. The subclades that compose each Ecological cluster are similar, suggesting that intra-subclade genomic variations may explain this distinct ecology.

It should be noted that a separate analysis of the MGII, derived predominantly from the UBA[22] dataset, was recently published by Rinke et al[40]. The two analyses provide a complementary analysis of MGII phylogeny, metabolic functional prediction, and global ecology. Importantly, both analyses provide a framework for MGII phylogenetic assignment that are directly comparable (Supplementary Figure 9). Based on normalized ranks derived from relative evolutionary divergence[41], Rinke et al. propose an Order-level classification for the MGII (*Candidatus* Poseidoniales) and subsequent classification of MGIIa and MGIIb in to the families *Ca.* Poseidoniaceae and *Ca.* Thalassarchaeaceae, respectively. Thus groups of MGII genomes designated as subclades in this analysis would correspond to genera in Rinke et al. There is a discrepancy in the number of genera identified by Rinke et al. ($n = 21$) and the 17 subclades described here. Five of the Rinke et al. genera (J1–3 and Q1–2) consisted of two or less genomes, and while counterparts are distinguishable amongst this analyzed dataset, the deep-branching nature of these genomes did not meet the criteria necessary for designations as a subclade (Supplementary Figure 9; Supplementary Discussion). The single genome representing genus L4 in Rinke et al. did not meet the criteria for inclusion in the non-redundant dataset, but is present in the same phylogenetic position for the full redundant dataset (Supplementary Figure 1). Further, Rinke et al. genera L1 and O3 was split in to two subclades each. This is likely the result of the specific cutoffs used to split clades O and L using relative evolutionary divergence[40] (Supplementary Figure 9; Supplementary Discussion).

Both analyses highlight the potential role of the MGII in particle degradation, specifically through the breakdown of proteins and fatty acids, particle attachment via flotillins, the presence of an archaeal flagellum operon in many subclades, and a varied distribution of green- and blue-light adapted proteorhodopsins among subclades. The horizontal gene transfer system proposed by Rinke et al. may be a possible explanation to account for the fine-scale differences observed within subclades detailed in this study. In addition to metabolic features discussed here, Rinke et al. discuss the lack of amino acid biosynthesis genes and the potential for mixed membrane synthesis among the MGII, which was not addressed here. Though the ecological approaches are not directly comparable, both analyses identify obligate mesopelagic MGII genomes.

There remain several important avenues of research that cannot be addressed despite the scale of the genome dataset. The MGII have previously been identified in filter fractions >3 μm and were hypothesized to have been attached to large plankton[9].

Virtually none of the *Tara* Oceans samples in the 3–5 μm range recruited to the MGII genomes, but the future release of metagenomes from the >5 μm size fraction, will allow for the screening of MGII lineages on larger particles. And while global metagenomes provide a metric for asking if an organism is present, it fails to capture what an organism is doing. Similarly, the release of the corresponding *Tara* Oceans microbial metatranscriptomes will allow for the assessment of what metabolisms are active in the MGII. With the identification of a putatively $Na^+$-driven ATP synthase, future research will be required to determine how the MGII ETC functions, especially with the prevalence of $H^+$-pumping proteorhodopsins. Should the MGII ETC obligately rely on $Na^+$ motive force, this may explain 'limited' distribution of MGII in marine environments, unlike the more global distribution of the *Thaumarchaeota* (formerly Marine Group I). The analysis of MGII genomic potential across 17 subclades allows for the reinterpretation of the role these organisms play in the cycling of HMW organic matter in the environment and opens these new avenues of research.

## Methods

**Genome selection and phylogenetic assessment**. MGII genomes that were publicly available prior to January 1, 2018[14,17,25–27] were collected from NCBI[42] and IMG[43] and were assessed using CheckM[44] (v1.0.11) to determine the approximate completeness and degree of a contamination (Supplementary Table 1). A 'Reference Set' of genomes that were >50% complete and <5% contaminated were included in downstream analysis, with the exception of two single-amplified genomes which were ~40% complete but possessed an annotated 16S rRNA gene sequence. Genomes with predicted phylogenetic placement within the MGII that were derived from the *Tara* Oceans metagenomic datasets[19,20,22,41] were collected and assessed with CheckM (as above). Genomes originating from Tully et al.[19,20] that had >5% predicted contamination were refined as described in Graham et al.[45]. Briefly, high contamination genomes originally binned with BinSanity[46] (v.0.2.6.2) had their sequences pooled with contigs from the same regional dataset (Tully et al.[20]) and were binned based on read coverage and DNA composition data using CON-COCT[47] (v.0.4.1; -c 800 -I 500). Putative bins determined by CONCOCT were identified that contained contigs from the BinSanity genomes. Overlapping bins and read coverage data from associated samples were profiled (anvi-profile), combined (anvi-merge), and visualized in Anvi'o[48] (v.3) (anvi-interactive). The putative MGII genomes were manually refined based on the hierarchically clustered coverage profiles and %G+C. The Anvi'o profiles used to perform this refinement for the analyzed genomes (see below) are available (https://doi.org/10.6084/m9.figshare.7154813.v1) and further details pertaining to the refinement method can be found in the Supplementary Methods.

A pairwise comparison of percent average nucleotide identity (ANI) was performed on the initial collection of redundant, putative MGII genomes ($n = 431$) to determine which genomes may represent identical genomic data (fastANI;[28] –fragLen 1500). Groups of genomes with ≥98.5% ANI were determined to be identical and the genome with the highest completion and lowest contamination estimates were selected as a non-redundant representative for further analysis (Supplementary Data 1; $n = 258$). Putative coding DNA sequence (CDS) were predicted for each using Prodigal[49] (v.2.6.3). The predicted proteins sequences for each genome were searched (HMMER[50] v.3.1b2; hmmsearch -E 1E-5) using HMM models representing the 120 single copy marker proteins as detailed by the Genome Taxonomy Database (GTDB v.86.0; http://gtdb.ecogenomic.org/) and sourced from TIGRfam[51] (v.15) and Pfam[52] (v.31.0). All proteins with a match to a single copy marker model were aligned using MUSCLE[53] (v.3.8.31; -maxiters 8) and automatically trimmed using trimAL[54] (v.1.2rev59; -automated1). Genomes with ≥60 single copy marker proteins were retained for further analysis ($n = 250$). Proteins alignments were concatenated and a phylogenomic tree was constructed using FastTreeDbl[55] (v.2.1.10; -gamma -lg; Supplementary Data 4 and 5). All described phylogenetic trees were visualized using the Interactive Tree of Life[56] (v.4.2.3). Pairwise amino acid identity (AAI) was calculated for the genomes, separated by into the major MGII clades (MGIIa and MGIIb) using CompareM (https://github.com/dparks1134/CompareM; v.0.0.23; aai_wf defaults; Supplementary Figure 2 and 3; Supplementary Data 2). Subclades 1–17 were defined based on a cutoff of 1.2 relative phylogenetic distance from the root and supported by clustering of pairwise relationships >70% ANI and >~70% AAI.

All genomes, including those identified as redundant[19–22,27], were assessed for the presence of the 16S rRNA gene using RNAmmer[57] (v.1.2; -S arch -m ssu; Supplementary Data 6). Identified sequences were combined with 16S rRNA gene sequences representing the available various reference genomes[14,17,25,26] and previously established clusters[11] (MGIIA clusters K, L, M; MGIIB clusters O, N, WHARN). As above, sequences were aligned using MUSCLE, and then manually trimmed to remove gaps in Geneious (v.6.1.8). A 430 bp alignment was used to construct a phylogenetic tree using FastTree (-nt -gtr -gamma; Supplementary

Figure 4; Supplementary Data 7). When possible, the previously defined 16S rRNA gene clusters were related back to the subclades identified on the single copy marker protein tree.

**Functional prediction**. A uniform functional annotation was applied to all predicted proteins for the set of analyzed genomes. Proteins were annotated with the KEGG database[58] using GhostKOALA[59] ('genus_prokaryotes + family_eukaryotes'; accessed September 17, 2018; v.2.0). Extracellular peptidases (enzymes predicted to degrade proteins) were identified with matches (hmmsearch -T 75) to PFAM HMM models[52] corresponding to MEROPS peptidase families[60] (Supplementary Table 4) that were predicted to have "extracellular" or "outer membrane" localization by PSortb[48] (v.3; -a) or an "unknown" localization with predicted translocation signal peptides by SignalP[61,62] (v.4.1; -t gram+). Carbohydrate-active enzymes (CAZy)[63] were identified (hmmsearch -T 75) using HMM models from dbCAN[64] (v.6). Functions of interest were predominantly identified based on the corresponding KEGG Orthology (KO) entry and GhostKOALA predictions. Specific functions of interest without a KO entry were searched using HMM models (hmmsearch -T 75) obtained from PFAM and TIGRFAM.

Predicted proteins of each genome were screened for matches to the rhodopsin PFAM model (PF01036; hmmsearch -T 75; Supplementary Data 8). In order to identify putative proteorhodopsins, sequences matching the rhodopsin HMM model were processed using the Galaxy-MICrhoDE workflow implemented on the Galaxy web server (http://usegalaxy.org) to assign rhodopsins to the MICrhoDE database[65]. The alignment generated from the workflow was manually trimmed to a 96 amino acid region conserved across all sequences, re-aligned using MUSCLE and used to construct a phylogenetic tree with FastTree (as above; Supplementary Data 9). The rhodopsins were predominantly assigned to three clades based on the phylogenetic relationships with other MICrhoDE sequences, unk-euryarch-HF70–59C08, unk-env8, and one unassigned clade. Two rhodopsins were assigned to additional clades, MICrhoDE clade IV-Proteo3-HF10_19P19 and another unassigned clade. Based on Pinhassi et al., unk-euryarch-HF70–59C08 and unk-env8 are also known as Archaea Clade-A and the unassigned clade belongs to Archaea Clade-B. A more detailed phylogenetic tree was constructed (as above) using only sequences from MGII and a reduced outgroup (Supplementary Figure 6). The MGII rhodopsin sequences were aligned using MUSCLE and were assessed for specific amino acids present at positions 97 and 108 to determine putative function and position 105 to determine putative spectral tuning (Supplementary Figure 5B).

The operon putatively encoding an archaeal flagellum was identified based on the presence of co-localized flagellar proteins FlaHIJ (K07331–3) and archaeal flagellins (PF01917). All genomes with possible co-localization of these proteins were identified (Supplementary Table 3). Putative operons from the analyzed TOBG genomes were visualized by subclade using the progressiveMauve aligner[66] (v.2.3.1; default) and longest contig containing the operon was selected to represent that subclade. Each representative was the compared to its phylogenetic neighbor using BLASTP[67] (v.2.2.30+; default parameters) to identify orthologs.

**MGII core genome analysis**. A pangenomic analysis was performed for the analyzed genomes using the Anvi'o pangenome workflow[68] (v.5). FASTA files for each genome were converted to a contigs database (anvi-script-FASTA-to-contigs-db) and each was amended with their corresponding GhostKOALA KEGG annotations (anvi-import-functions). The pangenome analysis (anvi-pan-genome) used the default parameters for minbit[69] (–minbit 0.5) and MCL[70] (–mcl-inflation 2) to generate protein clusters (PCs). Additional information regarding which clade and subclade each genome belonged was amended to the pangenome database (anvi-import-misc-data). Calculations to determine which functions were putatively enriched was performed at the clade (MGIIa or MGIIb) and subclade (1–17) levels (anvi-get-enriched-functions-per-pan-group). Enriched functions were determined to be a putative core function for the MGII if the corresponding protein cluster(s) was detected in ≥70% of all MGIIa and MGIIb genomes. Functions core to the MGIIa or MGIIb were protein cluster(s) present in ≥70% of genomes within the clade and in ≤30% of the genomes in the other clade. A function was determined to be putatively unique to a subclade if the protein cluster(s) was present in ≥70% of genomes within the subclade and in ≤10% of all other genomes (Supplementary Data 10).

**MGII relative fraction and environmental correlations**. The analyzed set of MGII genomes were used to recruit sequences from environmental metagenomic libraries, specifically 238 samples from *Tara* Oceans representing 62 stations and 118 samples from Ocean Sampling Day (OSD) 2014[38] (Supplementary Table 5 and 6). Metagenomic sequences were recruited using Bowtie2[65] (v.2.2.5; –no-unal). Resulting SAM files were sorted and converted to BAM files using SAMtools[71] (v.1.5; view; sort). BAM files were filtered using BamM (https://github.com/minillinim/BamM; v.1.7.3) to select for reads with ≥95% sequence identity with ≥75% alignment (–percentage_id 0.95 –percentage_aln 0.75). featureCounts[67,72] (v.1.5.0-p2; default parameters) implemented through Binsanity-profile[46] (v.0.2.6.4; default parameters) was used to generate read counts for each contig from the filtered BAM files. Read counts were used to calculate the relative fraction of each analyzed MGII genome in all metagenomic samples (reads recruited to a genome ÷ total reads in metagenomic sample) and reads per kbp of each genome per Mbp of each metagenomic sample (RPKM; (reads recruited to a genome ÷ (length of genome in bp ÷ 1,000)) ÷ (total bp in metagenome ÷ 1,000,000)) (Supplementary Data 11). Samples were divided into high (≥0.5% MGII recruitment) and low relative fraction samples (<0.5% MGII recruitment). Based on these designations, RPKM values for MGII genomes from *Tara* Oceans samples with high relative fraction with sufficient metadata (depth, temperature, and oxygen, chlorophyll, phosphate, and nitrate [measured as nitrate + nitrite]), were used in a canonical correspondence analysis[73] (CCA) in Past[74] (v.3.20). Due to the correlation of depth with a number of factors, temperature, chlorophyll, phosphate, and nitrate, depth measurements and one mesopelagic sample with all five parameters were removed from the final CCA. OSD samples consistently only collected temperature, distance from the coast, and salinity and were not included in CCA analysis. RPKM values for MGII genomes from high relative fraction samples were hierarchically clustered with average linkage clustering based on a Bray-Curtis distance matrix for both samples and taxon separately. Clustering was implemented with SciPy (http://www.scipy.org; v.1.0.0) and visualized with seaborn (http://seaborn.pydata.org; v.0.8.1). Ecological clusters of genomes were identified based on clades within each dendrogram, defined as <0.8 dissimilarity in the taxon dendrogram and <0.7 dissimilarity in the sample dendrogram, that corresponded to elevated RPKM values (Supplementary Figure 8). These ecological clusters were used to construct a reduced figure that maintained the sample and taxon clustering order (Fig. 5).

**Statistical analysis**. Average genome length and %G+C for the MIIa and MGIIb were compared using a two-sample, two-tail Student's *t*-test considering unequal variance.

Ecological clusters were correlated to temperature and oxygen, phosphate, and nitrate concentrations using a one-way PERMANOVA[75] implemented in Past3. PERMANOVA F-statistics were determined based on 9999 permutations performed on a Bray-Curtis[76] distance matrix. *p*-values were adjusted for multiple comparisons of related dependent variables using the Benjamini-Hochberg False Discovery Rate[77] correction. Past3 implements an ad hoc test of the pairwise PERMANOVA for all pairs of groups and corresponding *p*-values were adjusted with a Bonferroni correction (Supplementary Data 12). The PERMANOVA groups for the dependent variables were: temperature (3 groups), <10 °C, 10–20 °C, >20 °C; oxygen (4 groups), <110, 110–159, 160–200, >200 µmol/kg; phosphate (4 groups), below detection limit (BDL), <0.5, 0.5–2.0, >2.0 µM; nitrate (5 groups), BDL, <0.5, 0.5–1.9, 2.0–5.0, >5.0 µM.

**Reporting summary**. Further information on experimental design is available in the Nature Research Reporting Summary linked to this article.

## Data availability

The genomes used in this study are publicly available, except for a subset of deep-sea MGII from Guaymas basin presented in Li et al.[17] which were directly provided by the authors, and reference IDs are available in Supplementary Table 1. Additional files that are essential to the findings of this study, including the contigs and proteins used in analyses, phylogenetic marker sequences, the HMM models used to detected peptidases, the contigs with a putative motility operon, and the raw read counts for each OSD and *Tara* Oceans sample, are available through figshare (https://doi.org/10.6084/m9.figshare.7154813.v1). Genomes from Tully et al.[19,20] that were manually refined have been updated in NCBI with the corresponding accession IDs: NZKR02000000, NZKQ02000000, NZJY02000000, PAEM02000000, PADP02000000, PAUS02000000, PAMN02000000, PBGP02000000, PBGL02000000, NHGH02000000. A reporting summary for this article is available as a Supplementary Information file. The source data underlying Fig. 1, 2a–d, 3, 4, 5 and Supplementary Figures 1, 2, 3, 4, 6 and 8 are provided as a Source Data file.

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

## Acknowledgements

I would like to acknowledge and thank Drs. Rohan Sachdeva and Sarah Hu for reading, commenting, and enhancing early drafts of this manuscript. Elaina Graham provided invaluable support for running various bioinformatic pipelines and providing final edits. A special thanks to Dr. John Heidelberg for the suggestion of the (now defunct) original naming schema and acting as a combative reviewer. I would like to extend a special thank you to Dr. Johanna Holm for her patience when describing statistics and for proof-reading early drafts of the manuscript. I would like to thank Drs. Meng Li and Gregory Dick for prompt responses and access to the deep-sea MGII genomes. I would like to thank the Center for Dark Energy Biosphere Investigations (C-DEBI) for funding (OCE-0939654). And as I have noted before in previous research, I am grateful for the commitment of the *Tara* Oceans consortium to providing open access to their expansive metagenomic dataset and other researchers who have made their genomes so easily accessible. This is C-DEBI Contribution 452.

## Author contributions

B.J.T. designed and performed the research, analyzed the data, and wrote the paper.

## Additional information

**Competing interests:** The author declares no competing interests.

