## [Peer Review File · Nature Communications]

Editorial Note: This manuscript has been previously reviewed at another journal that is not operating a transparent peer review scheme. This document only contains reviewer comments and rebuttal letters for versions considered at Nature Communications .

Reviewers' comments:

Reviewer #1 (Remarks to the Author):

This manuscript provides a new analysis and proposes a classification for the important marine Archaea of group II. The manuscript has merit and provides interesting views about this important group. Unfortunately, this paper published recently has a similar proposal:

<https://www.nature.com/articles/s41396-018-0282-y>

To avoid confusion and also for scientific consistency, this manuscript needs to be rewritten considering the published proposal.

Reviewer #3 (Remarks to the Author):

The author has responded to all my comments, and has modified analyses (e.g., naming of lineages; phylogenomic analysis) and the text accordingly to all reviewers' requests. I believe the manuscript has significantly improved.

In light of the recent publication by Rinke et al. (<https://www.nature.com/articles/s41396-018-0282-y>), I consider that the study by Tully still represents an important contribution to the fields of microbial ecology and marine microbiology. In my view, the manuscript only requires few adjustments in order to take into account the recent publication. Especially, the linkage between 17 subclades by Tully and 21 genera by Rinke et al. needs to be made, maybe as a supplemental figure or table. This effort will create a bridge between the two studies, and in my view would benefit to the study by Tully, and by extension to our scientific community.

#Small points:

Ln 46: ref to properly format.

Ln 54: I suggest emphasizing that genomes are not redundant. It is a plus.

Ln 72: typo to fix in the sentence.

Ln 332: typo to fix in the sentence.

Ln 373: typo to fix in the sentence.

Reviewer #4 (Remarks to the Author):

In the revised manuscript "Metabolic Diversity within the Globally Abundant Marine Group II Euryarchaea Offers Insight into Ecological Patterns" Benjamin Tully convincingly describes the diversity and potential ecological role of MG II Euryarchaea. Most of my initial comments have been addressed to my satisfaction. The naming scheme has been updated and has gone away from the Lord of the Rings-based names (which I appreciate a lot, particularly as it might have caused legal issues regarding the copyright of the names). As a side note, the Hobbit-naming scheme is still mentioned in the Acknowledgements.

I am very much in favor of recommending this manuscript for publication, as it has undergone substantial improvements compared to what I have seen in the previous version. More robust statistics have been applied as well (e.g. PERMANOVA) and methods have now been described in a satisfying manner.

The only concern I see with the paper is, that in the meantime a similar story about MGII Archaea has already appeared in The ISME Journal (Rinke et al.). However, the story by Benjamin Tully has already been on BioRxiv for quite some time and Rinke et al. simply ignored Benjamin Tully's work. I think the current paper still deserves publication if a comparison to the Rinke et al paper is

included in the discussion section and the phylogeny and taxonomy are properly compared. A comparison of the potential metabolic traits would be necessary too. All these requests need, of course, to be in line with the journal policy about studies that get “scooped” while in peer review.

Minor comments:

- Abbreviations like RPKM should be introduced to the reader the first time they are being mentioned (e.g. Line 121)
- Legends and axis labelings of figures are sometimes too small to be seen without zooming in

Reviewer #1

This manuscript provides a new analysis and proposes a classification for the important marine Archaea of group II. The manuscript has merit and provides interesting views about this important group. Unfortunately, this paper published recently has a similar proposal:

<https://www.nature.com/articles/s41396-018-0282-y>

To avoid confusion and also for scientific consistency, this manuscript needs to be rewritten considering the published proposal.

Response: All three reviewers and the editor made this suggestion. Many of the core metabolic functions of the MGII are identified in both manuscript. Major differences in the manuscript stem from areas of interest unique to each author (Rinke *et al.* emphasized amino acid and membrane biosynthesis pathways, while I explored the role of Na⁺ motive force in the ETC and carbohydrate degradation. Rinke *et al.* emphasized a HGT-based theory for proteorhodopsin prevalence, a theory that is not in contradiction to the PR data presented in this manuscript). The phylogenetic trees presented in both manuscripts are virtually identical with only small differences affecting the number of subclades and genera present.

The following text has been added at the end of the Discussion:

It should be noted that during review a separate analysis of the MGII, derived predominantly from the UBA²² dataset, was published by Rinke *et al.*⁷¹. The two analyses provide a complementary analysis of MGII phylogeny, metabolic functional prediction, and global ecology. Importantly, both manuscripts provide a framework for MGII phylogenetic assignment that are directly comparable (Supplemental Figure 9). Based on normalized ranks derived from relative evolutionary divergence⁷², Rinke *et al.* propose an Order-level classification for the MGII (*Candidatus* Poseidoniales) and subsequent classification of MGIIa and MGIIb in to the families *Ca.* Poseidoniaceae and *Ca.* Thalassarchaeaceae, respectively. Thus groups of MGII genomes designated as subclades in this manuscript would correspond to genera in Rinke *et al.* There is a discrepancy in the number of genera identified by Rinke *et al.* (n = 21) and the 17 subclades described here. Five of the Rinke *et al.* genera (J1-3 and Q1-2) consisted of two or less genomes, and while counterparts are distinguishable amongst this analyzed dataset, the deep-branching nature of these genomes did not meet the criteria necessary for designations as a subclade (Supplemental Figure 9). The single genome representing genus L4 in Rinke *et al.* did not meet the criteria for inclusion in the non-redundant dataset, but is present in the same phylogenetic position for the full redundant dataset (Supplemental Figure 1). Further, Rinke *et al.* genera L1 and O3 was split in to two subclades each. This is likely the result of the specific cutoffs used to split clades O and L using relative evolutionary divergence⁷¹ (Supplemental Figure 9).

Both manuscripts highlight the potential role of the MGII in particle degradation, specifically through the breakdown of proteins and fatty acids, particle attachment via flotillins, the presence of an archaeal flagellum operon in many subclades, and a varied distribution of green- and blue-light adapted proteorhodopsins among subclades. The horizontal gene transfer system proposed by Rinke *et al.* may be a possible explanation to account for the fine-scale differences observed within subclades detailed in this study. In addition to metabolic features discussed here, Rinke *et al.* discuss the lack of amino acid biosynthesis genes and the potential for mixed membrane synthesis among the MGII, which was not addressed in this manuscript. Though the ecological approaches for the manuscripts are not directly comparable, both analyses identify obligate mesopelagic MGII genomes.

Methods for the Rinke *et al.* tree were add to the Supplemental Methods.

An additional discussion was added in the Supplemental Information for a slightly more detailed comparison of the trees. It reads:

On the Rinke *et al.* (2018) phylogenetic tree, the genera J1-3 and Q1-2 were represented by 1-2 genomes each. Two additional general level groups were missed by Rinke *et al.* due to the lack of inclusion of MGII genomes from Thrash *et al.* (2017). Both MGII Bin17 and MGII Bin15 may represent unique subclades, distinct from proposed clades J and Q. Subclade MGIIa.2 and genus K2 likely represent the same phylogenetic group, but have no overlapping genomes. Future phylogenetic trees combining the new genomes from Rinke *et al.* should alleviate this discrepancy. The single representative of genus L4 (UBA15) was present on the phylogenetic tree constructed for all redundant genomes for this manuscript. UBA15, detected by Rinke *et al.* in the Northwest Arabian Sea, had $\geq 98.5\%$ average nucleotide identity (ANI) to the genome SP339, reconstructed from the South Pacific, This expands the range of that genus beyond the Arabian Sea. Discrepancies between MGIIa.7 and MGIIa.8/genus L1 and MGIIb.12

and MGIIb.13/genus O3 could be resolved by slight modifications in the cutoff value for relative evolutionary distance applied to each genus. This change that would likely maintain the rank normalization groupings presented in Figure S5 in Rinke *et al.*

Reviewer #3

The author has responded to all my comments, and has modified analyses (e.g., naming of lineages; phylogenomic analysis) and the text accordingly to all reviewers' requests. I believe the manuscript has significantly improved.

In light of the recent publication by Rinke *et al.* (<https://www.nature.com/articles/s41396-018-0282-y>), I consider that the study by Tully still represents an important contribution to the fields of microbial ecology and marine microbiology. In my view, the manuscript only requires few adjustments in order to take into account the recent publication. Especially, the linkage between 17 subclades by Tully and 21 genera by Rinke *et al.* needs to be made, maybe as a supplemental figure or table. This effort will create a bridge between the two studies, and in my view would benefit to the study by Tully, and by extension to our scientific community.

Response: Please see the response to Reviewer #1.

#Small points:

Ln 46: ref to properly format.

Response: This formatting has been corrected.

Ln 54: I suggest emphasizing that genomes are not redundant. It is a plus.

Response: 'non-redundant' has been added to emphasize this aspect of the dataset.

Ln 72: typo to fix in the sentence.

Response: This typo has been corrected.

Ln 332: typo to fix in the sentence.

Response: This typo has been corrected.

Ln 373: typo to fix in the sentence.

Response: This typo has been corrected.

Reviewer #4

In the revised manuscript "Metabolic Diversity within the Globally Abundant Marine Group II Euryarchaea Offers Insight into Ecological Patterns" Benjamin Tully convincingly describes the diversity and potential ecological role of MG II Euryarchaea. Most of my initial comments have been addressed to my satisfaction. The naming scheme has been updated and has gone away from the Lord of the Rings-based names (which I appreciate a lot, particularly as it might have caused legal issues regarding the copyright of the names). As a side note, the Hobbit-naming scheme is still mentioned in the Acknowledgements.

I am very much in favor of recommending this manuscript for publication, as it has undergone substantial improvements compared to what I have seen in the previous version. More robust statistics have been applied as well (e.g. PERMANOVA) and methods have now been described in a satisfying manner.

The only concern I see with the paper is, that in the meantime a similar story about MGII Archaea has appeared in The ISME Journal (Rinke *et al.*). However, the story by Benjamin Tully has already been on BioRxiv for quite some time and Rinke *et al.* simply ignored Benjamin Tully's work. I think the current paper still deserves publication if a comparison to the Rinke *et al.* paper is included in the discussion section and the phylogeny and taxonomy are properly compared. A comparison of the potential metabolic traits would be necessary too. All these requests need, of course, to be in line with the journal policy about studies that get "scooped" while in peer review.

Response: Please see the response to Reviewer #1.

Minor comments:

- Abbreviations like RPKM should be introduced to the reader the first time they are being mentioned (e.g. Line 121)

Response: The abbreviation of RPKM in the Figure 2 legend has been explained.

- Legends and axis labelings of figures are sometimes too small to be seen without zooming in

I have made some adjustments to legends and axes for Figure 2, 3, and 5. I will work with the editorial team to make sure final figures are readable in the publication.

REVIEWERS' COMMENTS:

Reviewer #3 (Remarks to the Author):

In my view, the manuscript now provides an adequate link to the clades defined by Rinke et al. and does not require further changes in the text, besides small points I describe below.

-Please avoid the use of "manuscript" to describe this study, as well as the analysis done by Rinke. This includes the main text, but also the M&M section.

Also, in Ln 377 I would favor "It should be noted that a separate analysis of the MGII, derived predominantly from the UBA (ref) dataset, was recently published by Rinke et al. (ref)."

-It is unclear how sentences such as "most detailed view of these mesophilic archaea to-date" (abstract) will be viewed by the scientific community now that a complementary study has been published. I would recommend slightly tempering such sentences. "A comprehensive investigation of these mesophilic archaea", for instance, would still provide a strong statement, in my opinion.

-Ln 409: Many TARA Oceans metagenomes corresponding to size fractions >5micron have been generated and will be released soon (if they are not already available). It might be better to present this as a perspective rather than a limitation? In fact, these lineages will be screened for detection in larger size fractions in months/years to come.

I thank the author for efficiently responding to the various rounds of reviewing. I believe the manuscript has significantly improved.

Reviewer #3

In my view, the manuscript now provides an adequate link to the clades defined by Rinke et al. and does not require further changes in the text, besides small points I describe below.

-Please avoid the use of “manuscript” to describe this study, as well as the analysis done by Rinke. This includes the main text, but also the M&M section.

Response: The term ‘manuscript’ in the main body of the text has been removed. Five instances removed.

Ln. 239 ‘a subset of MGII genomes discussed in the manuscript in...’ is now ‘a subset of MGII genomes analyzed from...’

Ln. 373 ‘manuscripts’ is now ‘analyses’

Ln 378. ‘this manuscript’ is now ‘this analysis’

Ln 387. ‘Both manuscripts’ is now ‘Both analyses’

Ln 394. ‘which was not addressed in this manuscript’ is now ‘which was not addressed here’

Ln 394. ‘for the manuscripts’ was removed entirely. The sentence now reads “Though the ecological approaches are not directly comparable,...”

Also, in Ln 377 I would favor “It should be noted that a separate analysis of the MGII, derived predominantly from the UBA (ref) dataset, was recently published by Rinke et al. (ref).”

Response: The sentence has been updated to read as suggested.

-It is unclear how sentences such as “most detailed view of these mesophilic archaea to-date” (abstract) will be viewed by the scientific community now that a complementary study has been published. I would recommend slightly tempering such sentences. “A comprehensive investigation of these mesophilic archaea”, for instance, would still provide a strong statement, in my opinion.

Response: Agreed. The sentence has been updated to read as suggested.

-Ln 409: Many TARA Oceans metagenomes corresponding to size fractions >5micron have been generated and will be released soon (if they are not already available). It might be better to present this as a perspective rather than a limitation? In fact, these lineages will be screened for detection in larger size fractions in months/years to come.

Response: The sentence now reads: “Virtually none of the Tara Oceans samples in the 3-5µm range recruited to the MGII genomes, but the future release of metagenomes from the >5µm size fraction, will allow for the screening of MGII lineages on larger particles.”

I thank the author for efficiently responding to the various rounds of reviewing. I believe the manuscript has significantly improved.